# Food Allergen Nitration Enhances Safety and Efficacy of Oral Immunotherapy in Food Allergy

**DOI:** 10.3390/nu14071373

**Published:** 2022-03-25

**Authors:** Nazanin Samadi, Larissa Koidl, Martina Salzmann, Martina Klems, Natalie Komatitsch, Denise Schaffer, Eleonore Weidmann, Albert Duschl, Jutta Horejs-Hoeck, Eva Untersmayr

**Affiliations:** 1Institute of Pathophysiology and Allergy Research, Center of Pathophysiology, Infectiology and Immunology, Medical University of Vienna, 1090 Vienna, Austria; nazanin.samadi@meduniwien.ac.at (N.S.); larissa.koidl@meduniwien.ac.at (L.K.); martina.salzmann@meduniwien.ac.at (M.S.); martina.klems@oegk.at (M.K.); natalie.komatitsch@gmail.com (N.K.); d.schaffer@gmx.at (D.S.); eleonore.weidmann@gmx.at (E.W.); 2Department of Biosciences, University of Salzburg, 5020 Salzburg, Austria; albert.duschl@sbg.ac.at (A.D.); jutta.horejs-hoeck@plus.ac.at (J.H.-H.)

**Keywords:** oral immunotherapy, food allergy, protein nitration, modulated immune response

## Abstract

(1) Background: Posttranslational protein modifications have been demonstrated to change protein allergenicity. Previously, it was reported that pretreatment with highly nitrated food proteins induced a tolerogenic immune response in an experimental mouse model and in human immune cells. Here, we investigated a possible therapeutic effect of modified proteins and evaluated the safety of oral exposure to highly nitrated proteins in an experimental food allergy model. (2) Methods: BALB/c mice were orally sensitized towards ovalbumin (OVA) under gastric acid suppression. Thereafter, treatment via intragastric gavage with maximally nitrated OVA (nOVAmax) and OVA as a control was performed six times every 2 weeks. On the last day of experiments, all the treated mice were orally challenged with OVA. Systemic anaphylactic reaction was determined by measuring the core body temperature. Moreover, antibody levels, regulatory T cell numbers, cytokine levels and histology of antrum tissues were analyzed. (3) Results: After oral immunotherapy, OVA-specific IgE titers were decreased while IgG1 titers were significantly elevated in the mice receiving OVA. After oral challenge with OVA, nOVAmax-treated allergic animals showed no drop of the core body temperature, which was observed for OVA-allergic and OVA-treated allergic animals. Significantly fewer eosinophils and mast cells were found in the gastric mucosa of the allergic mice after nOVAmax treatment. (4) Conclusions: Oral immunotherapy with nOVAmax reduced allergic reactions upon allergen exposure and the number of allergen effector cells in the gastric mucosa. Thus, maximally nitrated allergens enabled an efficient and safe treatment for food allergy in our experimental model.

## 1. Introduction

Leonard Noon was the first to report about the application of allergen-specific immunotherapy (AIT) in 1911. He observed clinical improvements in grass pollen-allergic patients treated with grass extract injections [1]. His work was continued by his colleague John Freeman who published a larger observation study in Lancet in 1911 [2]. Noon and Freeman’s method was rapidly spread around the world by physicians trying to treat allergic diseases. In the first half of the 20th century, various types of allergen extracts, mainly pollen, were prepared and used in different studies. The first well-designed clinical trials with AIT were conducted in 1950 when Frankland et al. reported hyposensitization to be significantly more effective in higher doses for treatment of hay fever [3]. Over the years, AIT was developed from pollen extracts and whole bee venom extracts, chemically modified allergens (allergoids) and various recombinant allergens [4]. The goal of AIT is to induce desensitization and, if fully successful, tolerance. It is known to increase the amount of allergen-specific regulatory T cells (Treg), which suppress type 2 immune responses, to increase allergen-specific IgG4 antibody levels, to reduce mast cell and basophil activation [5,6] and to decrease allergen-specific IgE antibody levels [6]. A gradual increase in the administered allergen over time results in an elevated threshold of tolerated allergens while on therapy (known as desensitization) and should lead to clinical benefit even after discontinuation of treatment (known as tolerance or sustained unresponsiveness) [7]. Potential outcomes and markers of AIT were discussed in previous articles in detail (for overview, please see [8,9]).

Food allergies affect approximately 5% of adults and 8% of young children and has become a major global health concern [10]. In the past decades, remarkable advances have been made in the field of food-specific immunotherapy (IT), including oral immunotherapy (OIT), sublingual immunotherapy (SLIT) and, recently, epicutaneous immunotherapy (EPIT).

For OIT, allergenic food is mixed with a vehicle and is consumed in gradually increasing doses [5]. Protocols vary based on the type of food and vehicle used. All the studies have demonstrated that the majority of the participants treated with OIT can be successfully desensitized, while sustained unresponsiveness is less commonly achieved. Additionally, adverse reactions upon OIT are frequently observed [5].

In food allergy treatment, it is particularly concerning that most severe reactions occur unexpectedly at previously well-tolerated doses, being attributed to cofactors such as infection, exercise, anxiety or allergen co-exposure [11,12]. Maintained protection against allergic symptoms and safety have crucial implications for the future of OIT, highlighting the high need for further research efforts to define novel therapeutic approaches [13].

Modification of dietary proteins might be used to beneficially alter the immune response [14]. From previous studies, posttranslational protein modifications such as protein nitration are known to influence protein allergenicity. The route of exposure especially affects the elicited immune response as oral exposure results in a decreased sensitization capacity [15]. Moreover, recent results have revealed that oral exposure to nitrated food proteins efficiently modulates the immune response [16]. Additionally, the nitration degree influences the immunological impact as nOVAmax exposure prevents allergy development by elevated levels of regulatory T cells and decreased IgE, IgG1 and IgG2a titers. Upon pretreatment with nOVAmax, decreased expression of the activation marker CD86 and increased IL-10 levels in human monocyte-derived dendritic cells (moDC) were detected, while an enhanced number of proliferating memory Tregs derived from peripheral blood mononuclear cells (PBMCs) was observed [17]. Thus, the findings indicated that nOVAmax pretreatment induced a tolerogenic response in that experimental food allergy model as well as in human immune cells.

Here, we aimed to investigate the therapeutic potential of maximally nitrated food proteins as well as the safety of oral administration in an experimental food allergy model in mice.

## 2. Materials and Methods

### 2.1. Animals

Forty-seven female BALB/cAnNCrl mice (weight range: 15–20 g; provided with a health report certificate) were purchased from Charles River Laboratories (Germany) and housed under conventional conditions (12 h light and dark cycles, 22 °C ambient temperature) in a mouse facility at the Institute of Pathophysiology and Allergy Research, Medical University of Vienna. The mice were divided into six groups (groups A–E, *n* = 8 mice; group N, *n* = 7 mice) and kept in polycarbonate Makrolon type II cages (Ehret GmbH, Emmendingen, Germany) with aspen wood bedding (Ehret GmbH, Emmendingen, Germany) and filtered cage tops. The mice were provided with nesting materials and had access to food (egg- and cow milk-free diet, ssniff, Soest, Germany) and water ad libitum. After an acclimation period of 2 weeks, the experimental procedure was performed. The mice were treated according to the European Union guidelines of animal care and with permission of the Animal Ethics Committee of the Medical University of Vienna and the Austrian Federal Ministry of Science and Research (permission No. BMWF-66.009/0229-WF/V/3b/2017).

### 2.2. Nitration of OVA

To prepare ovalbumin (OVA) with a high nitration degree (nOVAmax), OVA (1 mg/mL; Sigma, Vienna, Austria) was dissolved in a Na_2_HPO_4_ buffer (10 mmol/L, pH 7.4) and cooked for 60 min at 100 °C. Afterwards, OVA proteins were mixed under continuous agitation with 0.5 mol/L tetranitromethane (TNM) in methanol (Merck, Darmstadt, Germany) at the molar ratio of TNM/tyrosine residues in OVA molecules of 10:1 in glass tubes for 60 min. The reaction was stopped by washing the samples thrice with a Na_2_HPO_4_ buffer using an Amicon Ultra-15 centrifugal filter device (Merck Millipore, Vienna, Austria) for 8 min at 4000 rpm with a 10 kDa cut-off membrane.

The resulting protein concentrations were measured with a Pierce BCA Protein Assay Kit (Thermo Fisher Scientific, Waltham, MA, USA) using OVA for the standard curve. To determine the number of nitrated tyrosine residues per OVA molecule, 3-nitrotyrosine (3-NT) (Sigma, Vienna, Austria) dissolved in 0.05 mol/L NaOH was used as the standard curve (range of the standard curve: 6.125–200 μmol/L). The protein samples were diluted 1:2 in 0.1 mol/L NaOH prior to measurement. The absorbance was measured at 428–650 nm with TECAN, infinite M200 PRO, and the number of 3-NT per molecule was calculated. The protocol yielded a nitration degree of 83.7% (10 tyrosine residues in OVA molecules).

### 2.3. Mouse Treatment Protocols

For food allergy induction, the mice (groups A, B and D) were gastric acid-suppressed by intravenous (i.v.) injection of 116 μg of the proton pump inhibitor (PPI) omeprazole (OMEP, Hexal, Germany) dissolved in 0.9% sodium chloride on three consecutive days every second week (Figure 1). On days 2 and 3 of each of the four sensitization cycles, the mice were fed with 200 μg OVA (Sigma Aldrich, St. Louis, MO, USA) mixed with sucralfate (Gerot Lannach Pharma Company, Lannach, Austria). Three groups of the mice remained naïve (C, E and N) (Table 1). 

Group A was subjected to final readout experiments immediately after immunizations. In order to evaluate anaphylactic reactions, an oral challenge with 2 mg OVA in 100 μL distilled water was performed. The core body temperature was measured before and 15, 30, 45 and 60 min after challenge. Thereafter, the mice were anaesthetized and exsanguinated by cardiac puncture. The spleens and intestinal lavages were harvested for further evaluations. From each animal gastric antrum, tissue was collected for histological evaluations.

In groups B–E, oral immunotherapy was performed with 200 µg nOVAmax (groups B and C) and 200 µg OVA (groups D and E) six times on three consecutive days every second week. Group N remained naïve for the entire procedure (Table 1 and Figure 1). To study the safety of OIT, we measured the core body temperature during the fourth and fifth rounds of therapy before and after 20 min oral gavage. To evaluate allergen-specific systemic responses after OIT with OVA, all the treated mice were orally challenged with 2 mg OVA, and the core body temperature was measured before and 15, 30, 45 and 60 min after challenge.

After the final anesthesia, blood was collected by cardiac puncture, and the spleens, intestinal lavages, gastric antrum and ileum tissues were harvested for further evaluations.

### 2.4. Evaluation of OVA-Specific Antibody Titers in Sera and Intestinal Lavages

The mouse serum samples were collected from the facial vein after immunization or cardiac puncture on the day of the final readout experiments and screened for OVA-specific IgE, IgG1 and IgG2a. ELISA plates (Bartelt, Austria) were coated with antibody standards and OVA (10 μg/mL) in a carbonate buffer (10 mM, pH 7.4) overnight at 4 °C. The serum samples were diluted 1:20 for IgE antibody detection and 1:100 for IgG1 and IgG2a detection in a dilution buffer (0.1% dried milk powder (DMP) in Tris-buffered saline and 0.5% Tween 20 (TBS-T) as previously described [18]. The small intestines were harvested and washed with 2 mL PBS with a protease inhibitor (Complete Mini, Roche, Basel, Switzerland). Thereafter, they were rotated at 4 °C for 4 h, and the supernatants were harvested. Afterwards, total IgA and OVA-specific IgA ELISA were performed as described [18].

### 2.5. Analysis of Murine CD4+ and Regulatory T Cells by Flow Cytometry

On the sacrifice day, 1 × 10^6^ splenocytes were stained for regulatory T cells (Tregs). For this, we used a mouse regulatory T cell staining kit (eBioscience, San Diego, CA, USA) and proceeded according to the manufacturer’s protocol. The number of the CD4+CD25+FOXP3+ cells was measured using FACS Canto II (BD Biosciences, San Jose, CA, USA), and the results were analyzed with the BD FACSDiva™ software (BD Biosciences, San Jose, CA, USA). Gating was performed as previously described [17].

### 2.6. Cytokine Measurements from Splenocyte Supernatants

The spleens of the mice were harvested and the splenocytes were further processed on the day of the final readout experiments. The cells were minced and passed through nylon cell strainers (40 μm, Corning Life Sciences BV, Amsterdam, The Netherlands). To lyse the erythrocytes, a lysis buffer (Lonza, Basel, Switzerland) was added for 5 min, followed by three washing steps with a medium (RPMI, 1% pen/strep, 1% L-glutamine and 10% fetal bovine serum (Thermo Fisher Scientific, Waltham, MA, USA). The splenocytes were counted with an automated Coulter Counter (TC10, Bio-Rad, Hercules, CA, USA), and 4 × 10^5^ cells/well were stimulated with 5 μg/mL of OVA or concanavalin A (ConA, positive control, Sigma, St. Louis, MO, USA) or medium (negative control) for 72 h at 37 °C. The supernatants were harvested, and the IL-4, IL-10 and IFN-γ levels were evaluated with ELISA kits (Thermo Fisher, Vienna, Austria).

### 2.7. Hematoxylin and Eosin Staining of Tissue Samples

Antrum tissues of the mice were dissected and embedded in the OCT compound on the day of the final readout experiments. The samples were stored at −80 °C until further use. With a cryostat (Cryostat Leica CM3050, Leica Biosystems, Wetzlar, Germany), 4 µm-thick cryosections were cut. After defrosting, the tissues were washed with PBS. For the staining itself, first, hematoxylin (1:1 in distilled water) was applied for 5 min, and afterwards, the samples were washed with tap water and PBS. Eosin Y (+0.05% acetic acid) was added for 1 min and washed away with distilled water. After the tissue samples were dehydrated using ascending ethanol series, they were incubated twice in N-butyl acetate (Sigma-Aldrich, St. Louis, MO, USA), mounted with Eukitt for preservation and covered with glass cover slips. Images of the stained sections were acquired with TissueFAXS (TissueFAXS version 4.2.6245.1019, TissueGnostics GmbH, Vienna, Austria), and they were analyzed with the HistoQuest image analysis software (HistoQuest version 6.0.1.125, TissueGnostics GmbH, Vienna, Austria). To evaluate the inflammation status of the tissue, five random areas of the tunica mucosa were chosen per sample and screened for immune cells. Accumulation of inflammatory cells and numbers of invading eosinophils were normalized to the evaluated areas.

### 2.8. Toluidine Blue Staining of Tissue Samples

To defrost the samples, the sections were kept at room temperature for 30 min. Subsequently, the slides were washed with PBS-T (with 0.1% Tween) for another 30 min, followed by a 5 min washing step with distilled water. Afterwards, each sample was incubated with a 0.1% toluidine blue solution (Sigma-Aldrich, St. Louis, MO, USA) for 5 min. After two additional washing steps with distilled water, the samples were dehydrated by an ascending ethanol series. After dehydration, the samples were incubated twice for 5 min each in N-butyl acetate (Sigma-Aldrich, St. Louis, MO, USA). Finally, the tissue samples were preserved with Eukitt (Sigma-Aldrich, St. Louis, MO, USA) and covered with glass cover slips (Thermo Fisher, Waltham, MA, USA).

### 2.9. Statistics

All the data were statistically analyzed using GraphPad Prism version 6 for Windows (GraphPad Software, San Diego, CA, USA). First, the data were checked for normal distribution using the Kolmogorov–Smirnov test. Depending on whether the data were normally distributed or not, the subsequent statistics were calculated using either the Kruskal–Wallis non-parametric test with Dunn’s multiple correction tests or by one-way ANOVA and Tukey’s post-hoc test; *p* values of < 0.05 were considered statistically significant.

## 3. Results

### 3.1. OIT Was Associated with Decreased OVA-Specific IgE while OVA-Specific Intestinal IgA Titers Remained Unchanged

Evaluations of the sera collected on the day of the final readout experiments indicated significant differences regarding OVA-specific antibody titers after oral immunotherapy (Figure 2B,E,H). Measurements of OVA-specific IgE titers after six cycles of immunization (Figure 2A) indicated significantly higher levels of IgE antibodies for the OVA-allergic mice groups (groups A, B and D), while after OIT (Figure 2B), OVA-specific IgE titers were significantly reduced in the nOVAmax- and OVA-treated mice (Figure 2A–C). 

In the naïve animals which were subsequently treated (groups C and E), the IgE levels remained at baseline after OIT comparable to the titers measured in the naïve animals (Figure 2B). Furthermore, all the groups were evaluated for systemic IgG1 and IgG2a levels. While OVA-specific IgG1 titers significantly increased only in the OVA-treated mice (Figure 2D–F), OVA-specific IgG2a titers slightly increased in all the treated groups during OIT (Figure 2G–I).

To evaluate the mucosal immune response, IgA titers were measured in the intestinal lavages. We observed significantly elevated total IgA levels in the allergic mice as well as in the OVA- and nOVAmax-treated mice (Figure 3A). Additionally, significantly higher OVA-specific IgA titers were found in the allergic mice and groups treated with OVA or nOVAmax compared to the naïve animals (Figure 3B).

### 3.2. Levels of Regulatory T Cells Are Comparable to Naïve Mice after Therapy

Flow cytometric analysis of T cells revealed comparable numbers of Tregs in the animals after OIT to those observed in the naïve animals (Table 2). The lowest level of Treg cells was observed in the allergic mice of group A. The allergic mice of groups B and D receiving OIT with nOVAmax or OVA revealed increasing numbers of Treg cells (Table 2).

### 3.3. OIT with nOVAmax Protected against Systemic Allergic Reactions after Oral Immunotherapy

In order to evaluate the safety of OIT, the core body temperature was measured before and 20 min after OIT gavage during the fourth and fifth rounds of OIT (Figure 4). No drop of body temperature was observed, indicating lack of an allergic response. For the readout of the clinical symptoms revealing efficacy of treatment, the animals were orally challenged with 2 mg OVA after six cycles of OIT, and the core body temperature was measured (Figure 5). All the allergic mice of group A showed a drop of body temperature after challenge. The allergic mice treated with OVA also revealed a significant drop of body temperature during 30 min after challenge (group D). The allergic mice treated with nOVAmax (group B), however, were protected against systemic allergic reactions during the entire period of follow-up.

### 3.4. Unchanged Cytokine Levels after OIT with nOVAmax

Cytokine release from the spleen cells stimulated with OVA was measured by ELISA. The IL-10 and IFN-γ levels were significantly higher only in the allergic mice (group A), while the nOVAmax- and OVA-treated mice indicated slightly elevated levels in comparison to the naïve mice (Figure 6A,B). The IL-4 levels released from splenocytes of the allergic mice were comparable to the values measured for the naïve group. Spleen cells of the treated allergic mice (groups B and D) released lower IL-4 levels (Figure 6C).

### 3.5. Reduced Allergy Effector Cell Influx Was Found in Gastric Mucosa of the nOVAmax-Treated Mice

On the day of the final readout experiments, gastric antrum tissue samples were collected from all the mice. Histological evaluations were performed with hematoxylin and eosin staining to study eosinophil influx and toluidine staining to count mast cells in order to determine signs of inflammation in the allergic mice (group A) and the OIT-treated allergic mice (Figure 7). Of interest, OIT with nOVAmax was associated with a reduced mast cell and eosinophil count comparable to the levels observed for the naïve animals. The allergic animals (group A) and the OVA-treated allergic mice (group D) revealed significantly higher numbers of allergy effector cells in the gastric mucosa (Figure 8).

## 4. Discussion

Even to date, the only commonly available treatment option in food allergies is avoidance of the triggering allergen. Since accidental exposure to food allergens is often inevitable and the prevalence of food allergies has increased over the recent decades, there is an urgent need to develop an efficient and safe treatment strategy. Exposure to nitrated food proteins was shown to induce a tolerogenic immune response [16]. Moreover, oral pretreatment with highly nitrated food proteins induced a tolerogenic response in an experimental mouse food allergy model as well as in human cells [17]. In the present study, we researched the potential therapeutic impact and safety profile of oral administration of maximally nitrated food proteins in an experimental mouse food allergy model.

Several studies indicated that OIT with low doses of food allergens such as eggs and hazelnuts can efficiently induce tolerance in allergic patients [19,20]. A recent study on peanut allergy in children indicated that OIT with the peanut allergen was associated with a reduction in peanut-specific IL-4, IL-5, IL-10, and IL-2 production by PBMCs compared to the placebo group, as well as a significant increase in peanut-specific IgG4 levels, with excellent safety profile [21]. This is in line with our own experimental data indicating that pretreatment with low dosages of food allergens induced a tolerogenic immune response in a mouse food allergy model [17].

Here, we performed OIT in allergic mice with low doses of nOVAmax and nonmodified OVA. In line with the previous results, we observed a significant reduction in IgE titers in the treated animals after low-dose OIT compared to the allergic group. Furthermore, we observed an increase in OVA-specific IgG2a, while after treatment, the IgG1 titers increased only in the OVA-treated allergic group.

In numerous studies, OIT was associated with increasing specific IgE levels during the rush phase and decreasing specific IgE levels during the maintenance phase [5]. Additionally, the IgG4 levels were elevated by OIT. IgG4 antibodies act as blocking antibodies to suppress IgE binding to its receptor on allergy effector cells and avoid degranulation, which is associated with protection against clinical symptoms [5,22]. In a murine model, OIT resulted in elevated allergen-specific IgE levels during therapy and in reduced allergen-specific IgE levels after treatment. Allergen-specific IgG2a titers increased after OIT, which indicated tolerance induction in mice [23].

In line with safety considerations for OIT [24], we measured the core body temperature before and after OIT with nOVAmax and OVA in mice during the fourth and fifth rounds of therapy. All the treated mice showed stable body temperature, which indicated safety of the treatment. Furthermore, the efficacy of OIT was evaluated by oral challenge with OVA and by measuring the core body temperature. Only the nOVAmax-treated allergic mice were protected against systemic reactions, showing no drop of body temperature, while the OVA-treated allergic mice showed a significant temperature reduction.

Despite short half-life (5–8 h) of IgE antibodies in serum, IgE can survive for months when bound to its high-affinity receptor (FcεRI) expressed on mast cells and basophils [25,26]. Cross-linking of FcεRI-bound IgE antibodies by allergens activates mast cells, leading to the release of preformed vasoactive amines and cytokines as well as de novo synthesis of proinflammatory lipids and chemokines associated with allergic reactions [27]. Upon mast cells activation, histamine inside the granules disassociates within minutes from the proteoglycans in the extracellular fluid by exchanging with sodium ions, resulting in smooth muscle contraction, endothelial cells and nerve ending activation and mucous secretion [28]. Not only mast cells, but also eosinophils are key players in gastrointestinal disorders, such as food allergy [29]. Eosinophils are inflammatory cells, and IL-5 release from T helper type II cells (Th2 cells) promotes their differentiation [28]. In murine models, the accumulation of eosinophils was reported to lead to airway inflammation, tissue remodeling and mucus secretion [30]. It was shown that the accumulation of eosinophils is observed in the gastric antrum mucosa of allergic mice [31,32]. Histological evaluations of gastric antrum mucosa indicated that only the nOVAmax-treated allergic mice had a significantly decreased number of allergy effector cells in the mucosal tissue, which was comparable to the levels observed for the naïve mice. The number of mast cells and eosinophils in the OVA-treated allergic group was similar to the allergic groups that had not undergone any type of oral immunotherapy.

T cells are also connected to the mechanisms of food allergy and food allergy immunotherapy in a crucial way. Whey-allergic mice showed not only reduced levels of Tregs and T helper 17 cells, but also reduced levels of H3 and/or H4 histone acetylation at relevant loci of both cell types. Whey-allergic mice also showed reduced H3 acetylation in T helper 1 cells, but the levels of the cells themselves were not reduced. In Th2 cells, no differences were observed in the study groups [33]. Therefore, that histone acetylation might not only play a mechanistic role in food allergy induction, but potentially also in food allergy therapy.

Here, we examined the therapeutic approach and safety of maximally nitrated allergens. Although generated in an experimental model, our data indicated that OIT with maximally nitrated proteins efficiently modulates an allergic immune response and can be considered as a safe allergen modification approach for future allergy treatment strategies.

## Figures and Tables

**Figure 1 nutrients-14-01373-f001:**
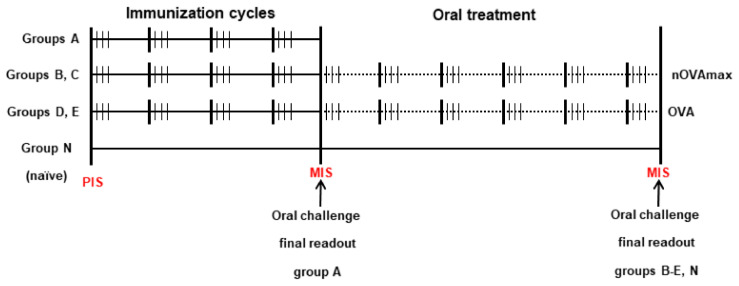
Mouse treatment protocol. One immunization/oral treatment cycle lasts 2 weeks. The bold lines in this figure indicate the start of a 2-week cycle. Immunization cycle: On days 1, 2 and 3, the mice of groups A, B and D received omeprazole intravenously. On days 2 and 3, the mice of the same groups were fed with OVA and sucralfate. Groups C, E and N did not undergo immunization. Oral treatment cycle: On days 1, 2 and 3 of each cycle, groups B and C received nOVAmax, groups D and E received OVA, groups A and N received no therapy. Abbreviations used: PIS—preimmunization serum; MIS—mouse immunization serum; nOVAmax—maximally nitrated ovalbumin; OVA—ovalbumin.

**Figure 2 nutrients-14-01373-f002:**
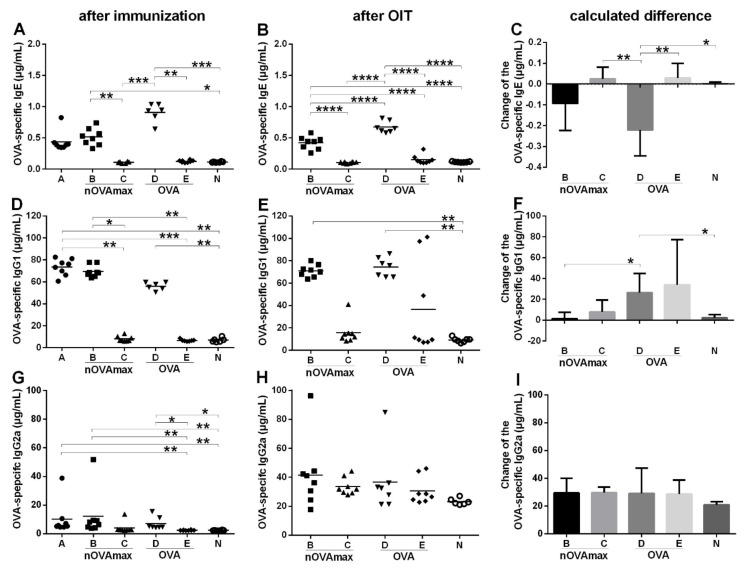
Evaluation of OVA-specific IgE, IgG1 and IgG2a titers in the mouse sera. The graphs (**A**,**D**,**G**) show the indicated serum antibody levels after all the four immunization cycles were finished. Graphs (**B**,**E**,**H**) show the serum antibody levels after oral immunotherapy. Graphs (**A**,**B**) show the IgE, (**D**,**E**) IgG1 and (**G**,**H**) IgG2a levels, respectively. Graphs (**C**,**F**,**I**) depict the calculated differences between before and after therapy. All the results were analyzed either with ANOVA combined with Tukey’s post-hoc test or with the Kruskal–Wallis non-parametric test with Dunn’s multiple correction. Statistically significant differences are indicated with brackets (* *p* < 0.05, ** *p* < 0.01, *** *p* < 0.001, **** *p* < 0.0001).

**Figure 3 nutrients-14-01373-f003:**
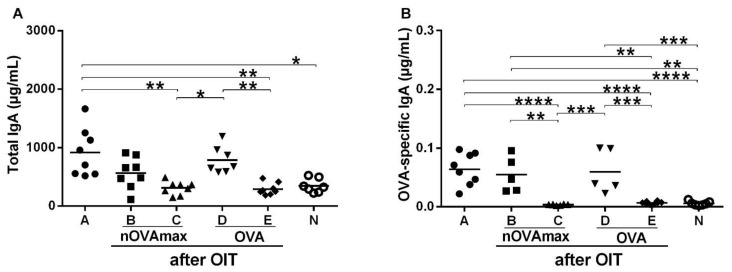
Measurement of OVA-specific IgA titers in the intestinal lavages after OIT. Intestinal lavages were collected and evaluated for (**A**) total and (**B**) OVA-specific IgA by ELISA. All the results for the total IgA were compared using the Kruskal–Wallis non-parametric test and Dunn’s multiple correction test, and the OVA-specific IgA results were analyzed with ANOVA combined with Tukey’s post-hoc test (* *p* < 0.05, ** *p* < 0.01, *** *p* < 0.001, **** *p* < 0.0001). Abbreviations used: nOVAmax—maximally nitrated ovalbumin; OVA—ovalbumin; OIT—oral immunotherapy.

**Figure 4 nutrients-14-01373-f004:**
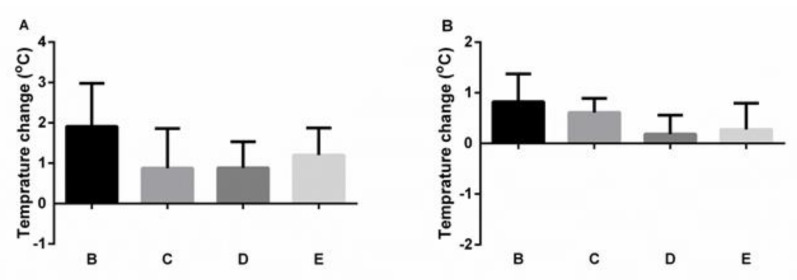
Core body temperature after OIT gavage. During the fourth and fifth rounds of OIT, before and 20 min after oral gavage of nOVAmax and OVA, the core body temperature was measured, and the temperature change is depicted. Graph (**A**) shows the values measured during the fourth round, graph (**B**)—the values measured during the fifth round.

**Figure 5 nutrients-14-01373-f005:**
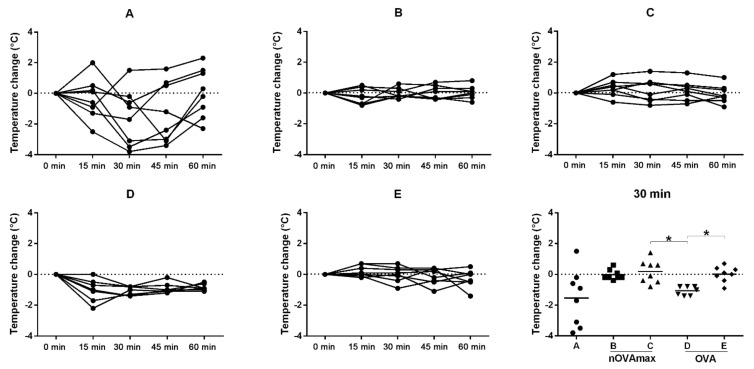
Measurement of the core body temperature for each mouse after oral challenge with 2 mg OVA. The baseline body temperature (0 min) of each mouse was measured prior to challenge, and body temperature was observed until 1 h after challenge (measurement timepoints: 15, 30, 45, 60 min after challenge). For each mouse, the temperature difference between the timepoints was calculated, and the changes in body temperature are shown in the graphs (**A**–**E**). The last graph illustrates all the measurements at the 30 min timepoint after oral challenge, and the values were compared using the Kruskal–Wallis non-parametric test with Dunn’s multiple correction (* *p* < 0.05).

**Figure 6 nutrients-14-01373-f006:**
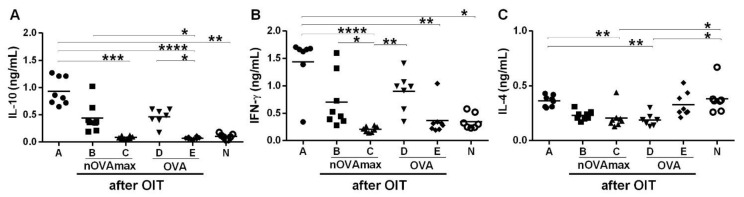
Cytokine release from the stimulated splenocytes. The spleens were isolated, and purified splenocytes were stimulated with 5 µg/mL of OVA. The supernatants were collected, and cytokine ELISAs were performed. Graphs (**A**–**C**) show the results for IL-10, IFN-γ and IL-4, respectively. The values were analyzed using the Kruskal–Wallis non-parametric test with Dunn’s multiple correction (* *p* < 0.05, ** *p* < 0.01, *** *p* < 0.001, **** *p* < 0.0001).

**Figure 7 nutrients-14-01373-f007:**
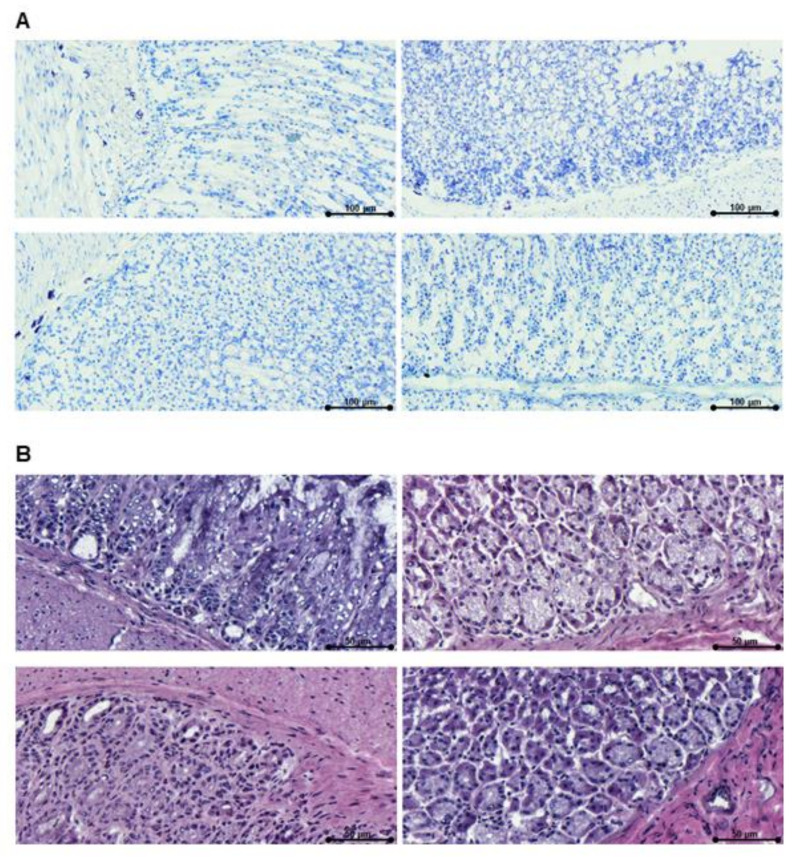
Murine gastric antrum tissues. Panel (**A**) shows mast cells in the gastric mucosa after staining with toluidine (bar size, 100 µm). Panel (**B**) depicts eosinophils after hematoxylin and eosin staining (bar size, 50 µm). In both panels, the figure on the upper left side shows representative staining from group A, and the figure on the upper right side—staining from group B. Lower left side and lower right side show stainings from group D and the naïve group, respectively.

**Figure 8 nutrients-14-01373-f008:**
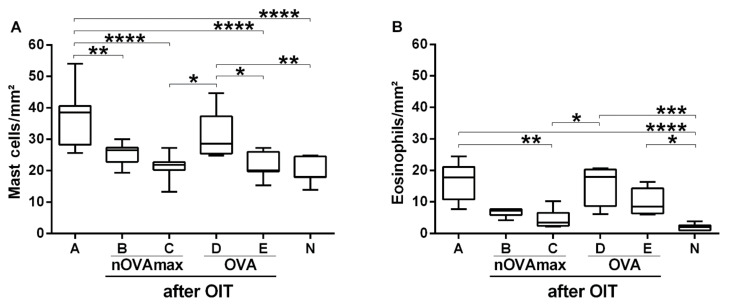
Number of mast cells and eosinophils per mm^2^ of gastric antrum tissues. The numbers of mast cells (**A**) and eosinophils (**B**) were normalized to the size of the evaluated tissue. All the results for mast cells were calculated with ANOVA combined with Tukey’s post-hoc test, for eosinophils—with the Kruskal–Wallis non-parametric test and Dunn’s multiple correction test (* *p* < 0.05, ** *p* < 0.01, *** *p* < 0.001, **** *p* < 0.0001).

**Table 1 nutrients-14-01373-t001:** Immunization protocol and oral immunotherapy. Overview of the substances used for immunization and/or therapy. Abbreviations used: nOVAmax—maximally nitrated ovalbumin; OVA—ovalbumin.

Group	Immunizations	Oral Immunotherapy
A	116 µg omeprazole, 2 mg sucralfate + 200 µg OVA	No therapy
B	116 µg omeprazole, 2 mg sucralfate + 200 µg OVA	200 µg nOVAmax
C	Naïve	200 µg nOVAmax
D	116 µg omeprazole, 2 mg sucralfate + 200 µg OVA	200 µg OVA
E	Naïve	200 µg OVA
N	Naïve	No therapy

**Table 2 nutrients-14-01373-t002:** Analysis of regulatory T cells by flow cytometry. Results of the statistical analysis of the flow cytometry experiment for each group are summarized in this table. Abbreviations used: SD—standard deviation.

Group	Mean	SD	*p*
A	3363	310.6	0.2
B	3400	445.1	0.2
C	3588	284.2	0.2
D	3600	262.7	0.2
E	3751	278.2	0.088
Naïve	3707	433.8	0.2

## Data Availability

The data presented in this study are available on request from the corresponding author.

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
