# Peer review of "Food Allergen Nitration Enhances Safety and Efficacy of Oral Immunotherapy in Food Allergy"

_nutrients, 2022, doi:10.3390/nu14071373_

Round 1

Reviewer 1 Report

With interest, I read the manuscript nutrients-1635374. It is a nice work based on a solid study and it is really difficult to find in it some failures or deficiencies.

My comments are thus minor and/or facultative in their character:

  1. Could the Authors think about any graphical abstract? Some illustration, maybe colorful could help to sell this article eve to the Readers not directly from the field.
  2. I do not see any reason why supplementary figures S1-S3 should be supplementary. Especially, Figure S1 should go to the main file, the best somewhere close to Table 1.
  3. The legend to Figure S1 should be, however, more elaborated.
  4. Generally, abbreviations used in the figures or tables should be explained in the legends/footnotes, e.g. “PIS” or “MIS” in Figure S1.
  5. Lines 325-327. Some reference support would be useful (PMID: 22909159, 17145160).
  6. The Discussion is a bit ascetic and could be expanded. It is clear from the manuscript that T cells are crucial in the mechanisms of food allergy and its immunotherapy. Their differentiation is strictly epigenetically mediated and the contribution of some epigenetic mechanisms such as histone acetylation to the development of food allergy has been reported (PMID: 33086571). Could epigenetic changes play a role also in the effects observed by you here?
  7. Could DNA methylation contribute as well and could it be a marked of the successful treatment (PMID: 29779209)?

Reviewer 2 Report

nutrients- 1635374

Comments and Suggestions for Authors

The paper by Samadi et al. examines the role of allergenic protein nitration as a method of immunoreactivity reduction. Nitrated OVA was applied as a therapeutic agent for OIT.

The issue is very important however some aspects need to be verified.

The manuscript has been prepared with care, however, a few issues are treated briefly and requires supplementation.

  1. Why the authors chose this type of intragastric immunization. It is visible in IL-4 content in medium after spleen incubation that the immunization was not permanent. Also specific IgE in blood seems not so significant?
  2. Why did they not administer intraperitoneal immunization which improves efficacy especially in the BALB / c mouse model?
  3. Which mouse strain was selected (pure BALB / c or some modification).
  4. How many times the experiment was repeated?
  5. Why the specific IgA turnover has not been measured. It should be presented both sIgA in blood and intestinal compartment?
  6. Was there applied any adjuvant? If ‘not’ why?
  7. How was performed the gating in cytometric analysis? How many counts were taken for analysis.
  8. The referred changes in the antibody levels do not appear to be significant. A lot of significance appears in the graphs. Although it is known that in studies on animal models, individual variability is high, but precisely because of the values shown for individuals, it is hard to believe that the analysis gave so many highly significant differences.

In reviewers opinion there are also some less important things but also requiring correction eg:

  • In the Table 1 it is no info about the application of OVA for the sensitization. It looks lie th mice model was immunized with 116 µg omeprazole + 2 mg sucralfate not with antigen.
  • There is difference between post-tests and post –hoc tests. In the context of statistics it is post-hoc.
  • In the introduction, it would be more useful to explain what are the parameters deciding about the effectiveness of specific immunotherapy and how it was considered if the therapy is permanent or temporary rather than the providing the story of who and when invented it. Nevertheless, it is an interesting way to introduce
  • I understand that the manuscript was prepared for a different journal and required a limited number of references and signs, but Nutrients do not impose such a limitation. I would suggest discussing the results in more detail.
  • Line 300- ‘peanut-specific IL-4, IL-5, IL-10, and IL-2 production’- I understand this is a quotation but what do the authors have in mind regarding cytokine specificity for antigen
  • The publication is written in such a way as if the authors themselves were not convinced that the conclusions were correct.

For the reviewer the indicated points in the work require explanations for further proceedings.

Reviewer 3 Report

In this paper the authors study nitrated food poteins in immunogenicity, trying to reduce the allergic response. The authors conclude that oral immunotherapy with nOVAmax reduced allergic reactions by reducing the number of effector cells in the gastric mucosa.

Table 1 and 2 carry only the title without any explanation.

The legends of the figures do not sufficiently explain the content.

A blot on the results would be needed to confirm the results and also for a better presentation.

  • This original paper looks good to me, but it should be better presented. For example it lacks some parts. To make this paper more interesting for the readers of this important journal, the authors, in relation to their data, should talk about intestinal and respiratory dysbiosis and the probiotics. In this regard, below I report 2 interesting articles that should be studied, incorporate their meaning and report them briefly in the discussion and in the list of references. Very recently, on these topics, 3 interesting articles have been published, which I suggest to read them incorporate their meaning and report them in the discussion and in the list of references.

A probiotic mixture in patients with upper respiratory diseases: the point of view of the otorhinolaringologist.

Gelardi M, La Mantia I, Drago L, et al. J Biol Regul Homeost Agents. 2020 Nov-Dec;34(6 Suppl. 1):5-10.

Probiotics in the add-on treatment of laryngotracheitis: a clinical experience.

La Mantia I, Gelardi M, Drago L, et al. J Biol Regul Homeost Agents. 2020 Nov-Dec;34(6 Suppl. 1):35-40.

I believe these suggestions are important for improving this paper.

Round 2

Reviewer 2 Report

Provided answers and cited publications complete the topic and dispel doubts.